# Histopathological Analysis of Selected Organs of Rats with Congenital Babesiosis Caused by *Babesia microti*

**DOI:** 10.3390/vetsci10040291

**Published:** 2023-04-14

**Authors:** Krzysztof Jasik, Anna Kleczka, Sandra Filipowska

**Affiliations:** Department of Pathology, Faculty of Pharmaceutical Sciences in Sosnowiec, Medical University of Silesia in Katowice, Ostrogórska 30, 41-200 Sosnowiec, Poland

**Keywords:** *Babesia microti*, babesiosis, tick-borne diseases, histology, kidney, spleen, transmission electron microscopy

## Abstract

**Simple Summary:**

Babesiosis is a rare zoonotic disease caused by protozoa of the genus *Babesia*. Humans most often become infected with babesiosis as a result of the transmission of the protozoan by an infected tick. Blood transfusions, transplanted organs, and vertical transplacental transmission from the mother to the fetus are much rarer (but possible) ways of transmission of the pathogen. The main danger of this disease is the multiplication of the protozoan in the erythrocytes, which leads to their damage and disintegration. The main effect of the infection is anemia and symptoms of hemolysis, but in some people—especially those suffering from impaired immunity—multi-organ failure and even death can occur. In this study, the effect of the transplacental transmission of *Babesia microti* on the spleen and kidneys of rats was investigated. It has been shown that congenital babesiosis can damage these organs and impair their functioning. In this work, we try to convince that the possibility of intra-uterine infection of the fetus should lead to the introduction of mandatory screening tests for carriers of pathogenic species of the genus *Babesia*, especially in young women, blood donors, and people exposed to tick bites for occupational reasons.

**Abstract:**

Babesiosis is a tick-borne disease with an increasing number of cases each year. Due to the non-specific symptoms of babesiosis, insightful analyses of the pathogenesis of babesiosis are still very important. Transmission of the disease occurs in a few ways, which makes laboratory diagnosis of piroplasmosis important. Complications associated with the infection can be tragic, especially in patients with immunological disorders. The aim of this study was the histopathological analysis of the spleen and kidney of young Wistar rats infected transplacentally with *Babesia microti*. Female rats were infected with a reference strain of *B. microti* (ATCC 30221), and then, birth 3-week-old males were euthanized with isoflurane. Subsequently, the material was collected at autopsy for microscopic and ultrastructural examination. Microscopic and ultrastructural analysis of the spleen and kidney showed degenerative changes within the organ parenchyma and the capsules surrounding the organ. Regenerative and reparative changes through mitotic divisions of parenchymal cells were also evident. Merozoites of *B. microti* were visible in the section of erythrocytes and the cells building the organ stroma. The results presented in this study proved the negative effects of *B. microti* on cells and tissues in rats with congenital babesiosis.

## 1. Introduction

Human babesiosis, also known as “northern malaria”, is a tick-borne disease caused by an intraerythrocytic protozoan. The vector of the pathogen and its definitive host are ticks of the genus *Ixodes*, mainly *Ixodes ricinus, Ixodes persulcatus,* and *Ixodes scapularis* [1]. Among over 100 species of the genus *Babesia*, the most common species causing disease in humans are *Babesia microti* and *Babesia divergens* [2]. Babesiosis is one of the most frequently reported tick-borne diseases next to tick-borne encephalitis (TBE) and Lyme borreliosis. It is included in the group of emerging tick-borne diseases due to the significant increase in cases in recent years [3].

The natural hosts of *Babesia microti* are *Peromyscus leucopus* [4]. However, the infection of the other rodents with *Babesia microti* was also observed.

The phenomenon of congenital babesiosis was observed in various rodents, with the prevalence of transplacental infection not being the same. For example, it was shown to be 98% in *M. arvalis* and 46% in *M. oeconomus* [5].

Research on vertical transmission of *Babesia microti* performed on Balb/c mice showed that acute *B. microti* infection prevents pregnancy initiation and embryonic development in the first trimester and causes severe complications, while chronic *B. microti* infection has no detrimental effect on pregnancy initiation and development but causes congenital complications [6,7].

The course of experimental infection by the *B. microti* (Franca) Reichenow ATCC 30221 strain in laboratory rats has been studied in our department for many years. We found that Wistar rats succumbed to infection by this strain of *B. microti*, which caused quite a few changes in the organs of these animals [8,9].

A considerable prevalence of vertical transmission in rats of the pathogens used in the experiment was observed [10,11].

*Babesia microti* (Franca) Reichenow ATCC 30221 is a strain of a parasitic protozoan that was isolated from a male with piroplasms from Nantucket Island, Massachusetts. Since the research presented here is the basis for further studies on babesiosis in humans, a strain of *B. microti* that can potentially infect humans was used.

Human babesiosis is most often transmitted through the bite of an infected tick and less often through organ transplants, infected blood, and blood product, or transplacental transmission, which is the cause of congenital babesiosis [12,13]. Congenital babesiosis occurs in infants who are infected by the transplacental or perinatal route. *Babesia* spp. parasites likely cross the placenta during pregnancy or delivery to the baby [14].

Most infections in immunocompetent people are asymptomatic or mildly symptomatic, while in elderly and immunocompromised patients, the disease can be acute or fatal [15,16,17]. Penetration of erythrocytes by *Babesia* spp. leads to symptoms, such as fever, chills, and hemolytic anemia, so the symptoms of babesiosis can be confused with those of malaria. A severe form of the disease can lead to serious complications that threaten the health and life of the patient. Complications of severe babesiosis can lead to respiratory failure, disseminated intravascular coagulation, congestive heart failure, and renal or hepatic failure. A dangerous complication for life is a heart attack or rupture of the spleen. Moreover *Babesia* spp. is a parasitic agent mimicking the HELLP syndrome (hemolysis, elevated liver enzymes, and low platelets) characterized by hemolysis, elevated liver enzymes, and thrombocytopenia. Recognizing and treating this condition is important to respond quickly and start appropriate treatment. Cases of congenital babesiosis were detected 19 to 41 days after birth and were characterized by asymptomatic maternal infection or fever, neonatal hemolytic anemia, and thrombocytopenia. The treatment, in this case, is antibiotic therapy and a blood transfusion [18].

The diagnosis of babesiosis should be based on a detailed history, including clinical symptoms, history of travel to or living in endemic areas, tick bites, recent blood transfusions, and tick-bitten pregnant women. Further analysis includes laboratory diagnostic methods to confirm infection with *Babesia* spp. [19].

Microscopic and ultrastructural analyses of selected organs of rats with congenital babesiosis are the basis for a thorough understanding of the etiopathogenesis of the disease, as well as for understanding the mechanism of pathological changes. In the future, these observations may allow for the improvment in the diagnosis of the disease before the damage to the main organs occurs. In addition, this research could improve methods of laboratory diagnosis, effective treatment, as well as prevention against infection. It is also an important goal to document the pathological changes in babesiosis because, from a clinical point of view, this disease in humans is marginally treated in Europe.

## 2. Materials and Methods

### 2.1. Breeding Conditions

The biological material was obtained from 8 female Wistar rats, which were divided into 2 groups—4 individuals constituted the study group and the 4 rats were classified into the control group. The conducted experiment was supported by the consent of the Local Ethical Committee for Animal Experiments No. 109/2013 with subsequent annexes extending the research group.

Animal breeding was carried out at the Center for Experimental Medicine of the Medical University of Silesia in Katowice. The males were bred individually in standard cages with optimal zoohygienic conditions (i.e., constant air temperature (20–22 °C), humidity 50–60%, and an equal daily cycle (12 h light and 12h dark) with unlimited access to food and water. The test group was inoculated intraperitoneally with 0.5 mL of a suspension of the reference strain *B. microti* (Franca) Reichenow ATCC 30221 (ATCC, Manassas, VA, USA) containing about 5 × 10^6^ piroplasms. The *Babesia microti* strain used in the study was originally isolated from humans. Females from the control group were inoculated intraperitoneally with saline at a dose of 0.5 mL.

### 2.2. Parasitemia Control, Euthanasia of Rats, and Collection of Material for Research

After the animals were inoculated, the development of parasitemia was observed. Methanol-fixed and May-Grünwald-Giemsa (POCh, Gliwice, Poland) stained smears of peripheral blood collected from the tail vena cava were analyzed. Smears were observed in an Olympus BX60 light microscope magnified 2200 times. After staining and drying, the smears were sealed in Mounting Medium DPX synthetic resin (POCh, Gliwice, Poland). A peak of parasitemia was observed 3 weeks after the inoculation of the rats. During this period, the females were mated using the harem method. Birth rats were tested for *B. microti* after 3 weeks. Verification of parasitemia was completed in a double step. It was first analyzed by MGG blood staining and later by using the FISH method.

In the third week of postnatal development, transplacental infected male rats and male rats of the control groups were euthanized. The procedure was performed in a sealed desiccator with an overdose of isoflurane (Fo-rane^®^—Baxter, Deerfield, IL, USA), and the organs were removed for histological and ultrastructural examination.

Before the histological investigation, the presence of *B. microti* in the examined tissues was checked with the FISH method. The peripheral blood smears for the FISH technique were fixed in a 2.5% paraformaldehyde solution. Detection of *B. microti* genetic material in tissue sections obtained using a cryostat and blood smears was performed using a commercially available kit, i.e., Histology FISH Accessory Kit (DAKO, Carpinteria, CA, USA). In this study, a fluorescein-labeled probe complementary to the fragment of *B. microti* 18S rDNA gene was used. The probe sequence was as follows: 5′-fluorescein-GCCACGCGAAAACGCGCCTCGAfluorescein-3′ [20]. Oligonucleotide synthesis was performed at the Institute of Biochemistry and Biophysics in Warsaw.

### 2.3. Histological and Ultrastructural Studies

The collected tissues of the study animals were analyzed using routine histological and ultrastructural techniques. Additionally, the presence of *Babesia microti* was analyzed in the investigated tissues [8].

Tissue fragments collected during the autopsy were fixed for 5 days at room temperature in Bouin’s solution (a mixture of picric acid, buffered formalin, and glacial acetic acid mixed in a volume ratio of 15:5:1). The fixed tissues were rinsed with 80% ethyl alcohol for 7 days, then dehydrated in an alcohol series, X-rayed in benzene, and embedded in paraffin. The samples embedded in paraffin were cut into 6 μm thick sections using a microtome. The obtained preparations were stained with the three-color AZAN method. Dehydrated sections were coverslipped using DPX Mounting Medium (POCh, Gliwice, Poland). The finished preparations were observed using an Olympus type BX60 light microscope.

The collected tissues were fixed with Karnowski’s solution (2.5% mixture of glutaraldehyde and paraformaldehyde solution in a 1:1 ratio in phosphate buffer). The material was then rinsed with phosphate buffer and re-fixed in 1% osmium tetroxide solution in the same buffer. Following fixing and rinsing in the buffer, the tissues were dehydrated in ethanol and propylene oxide series and then embedded in Poly/Bed^®^ 812 Embedding Media/DMP-30 Kit epoxy resin (Polyscience, Inc., Warrington, PA, USA). Using an ultra-microtome (Leica ultracut UCT), the material was cut to obtain semi-thin sections, 0.5 µm thick, which were stained at a temperature of approx. 80°C with methylene blue (AppliedChem, Darmstad, Germany). Obtained semi-thin sections were used for the approximate assessment of cross-sections and selection of places from which ultra-thin sections were obtained, intended for analysis in a transmission electron microscope. These sections were obtained using a diamond knife (DIAMOND Switzerland). Ultrathin sections with a thickness of 70–80 nm were placed from a knife onto copper 300 mesh grids. After drying, the sections on the grids were contrasted with a solution of uranyl acetate and lead citrate solution.

The ultrastructural observations were performed by transmission electron microscope FEI Tecnai G2 BioSpirit at an accelerating voltage of 120 kV.

Electron microscopic examinations were performed in the Electron Microscopy Laboratory, Department of Histology and Cell Pathology in Zabrze, Medical University of Silesia in Katowice.

## 3. Results

### 3.1. Verification of the Presence of the Merozoites Babesia microti

In blood smears collected from the rats examined with congenital babesiosis, abundant forms of trophozoites of *B. microti* were observed. They appeared in both MGG-stained and FISH-stained slides (Figure 1A,B).

DNA labeling of *B. microti* in kidney and spleen sections has proven the presence of *B. microti* merozoites in these tissues (Figure 2A,B).

### 3.2. Histological Structure of the Kidneys of Rats with Congenital Babesiosis

The rat’s kidneys are smooth, reddish brown, and contain over 300 million nephrons. Histological sections of the kidneys of rats with congenital babesiosis showed typical layers, i.e., fibrous capsule and cortical and medullary parts (Figure 3). The anatomy was generally not malformed. In some places, the tubules or spaces between the tubules at the border of the cortical and medullary parts were widened (Figure 4). Degenerative changes and deposits were evident in the proximal and distal tubules. Using higher magnifications, *B. microti* merozoites were noticeable in sections of tubular cells and endothelial cells. In some cross-sections through the tubules, erythrocytes were observed, and the vessels between the tubules showed features of hyperemia. A large number of tubular cells showed signs of degeneration, discontinuity of cell membranes, and vacuolation (Figure 4 and Figure 5).

### 3.3. Ultrastructure of the Kidneys of Rats with Congenital Babesiosis

In cross-sections through the kidneys, similar destructive changes were observed in tubular epithelial cells. Numerous peroxisomes were noted in many cells of proximal tubules throughout the cytoplasm (Figure 6A). The cytoplasm of some tubular epithelial cells was markedly altered. Dispersed cytoplasm with poor osmophilicity was observed in them (Figure 6B). The basal labyrinth was distorted and there were no mitochondria between the membrane folds (Figure 7A). Observations of the ultrastructure of the matrix space of the renal cortex revealed bundles of collagen fibers between the tubules. The basement membranes of some tubules were markedly thickened (Figure 6B and Figure 7A). Within these membranes, osmophilic inclusions and lighter spaces were visible. This membrane had a heterogeneous structure. In some of the observed areas, the destruction of the basement membranes and their lack of continuity was noted. Mitochondria extended beyond the basal labyrinth and had the correct orthodox structure. In the cytoplasm, a large number of lysosomes with a characteristic electron-dense content were observed. The renal corpuscles were properly structured, i.e., the correct arrangement of podocytes and their protrusions, as well as the typical fenestrated structure of the endothelium. A slight increase in the amount of mesangial substance was noticeable in places. Dilatations of the urinary space were more characteristic, especially in the marginal parts of the corpuscles, between the glomerulus and the Bowman’s capsule (Figure 7B).

### 3.4. Histological Structure of the Spleen of Rats with Congenital Babesiosis

The spleen develops from mesenchymal cells that divide and differentiate to form the connective tissue skeleton of the spleen. In rats, it is a peripheral lymphoid organ weighing 750–1350 mg. The spleen consists of three main zones, i.e., the hematogenic red parenchyma, the lymphoid white parenchyma, and the marginal zone. It is surrounded by a relatively thin connective tissue capsule composed of collagen and elastin fibers, smooth myocytes, and unmyelinated nerve fibers. The stroma of the spleen is formed by reticular connective tissue proper with reticular fibers and a few fibroblasts, and white and red pulp are distinguished in the parenchyma [21].

In the presented studies, in the histological structure of the spleen of intrauterine infected rats with *B. microti*, no clear deviations from the normal architecture of the organ were observed, appropriate to the young age of the observed individuals. Both white and red pulp were visible. However, the line between them was blurred (Figure 8). The lymphoid follicles were irregular in shape and the marginal layer was barely perceptible. The fibrous capsule of the spleen of rats under physiological conditions is characterized by a small amount of connective tissue. In the observed preparations of spleen sections taken from rats with congenital babesiosis, a small amount of it was visible with the presence of mesothelial cells covering its surface. The connective tissue capsule detached from the parenchyma of the organ in places, creating fissured spaces. These dissections indicate degenerative changes, possibly associated with edema. The trabeculae extending from the splenic capsule were visible and contained vessels with fibrin-precipitated hyperemia and numerous blood cells adhering to the endothelial wall (Figure 9). In addition, a characteristic feature was the differentiation of erythrocytes.

Using high magnification, the presence of intracellular inclusions in the red pulp could be seen, which, based on the structure, can be defined as invasive forms of *B. microti*. In some preparations, mitotic cell divisions were also observed, which are a manifestation of repair and regeneration processes or, at this stage of development, still in the residual phase of erythropoiesis. Voids were observed in many areas, indicating atrophic retrograde changes, as well as macrophages responsible for the phagocytosis of disintegrating erythrocytes and *B. microti* protozoa (Figure 10).

### 3.5. Ultrastructure of the Spleen of Rats with Congenital Babesiosis

The results of the analyses of the ultrastructure of the spleen of rats with congenital babesiosis caused by *B. microti* confirmed the studies at the light microscope level. White and red pulp were observed, although the marginal zones were difficult to identify. The white pulp of the spleen was characterized by the presence of cells of variable structure and size. Within this zone, there were cells with cytoplasm of different electron densities (Figure 11 and Figure 12). Granular cells with a neutrophil structure were observed (Figure 11A). Most of the cells had normal, orthodox mitochondria, although diffuse areas indicated degenerative changes in some cells (Figure 11A,B and Figure 12A). In the residual erythrocytes, apart from the white matter, membraneous inclusions were observed, which were sections through the merozoites of *B. microti* (Figure 11A and Figure 12B).

*B. microti* trophozoites were also observed outside the blood cells, in the cells of the splenic parenchyma. Cells in the red matter showed high activity and divided mitotically (Figure 11B). The red pulp of the spleen contained numerous macrophages in the cytoplasm of which heterophagosomes were present. Inside these structures, there were fragments of digested, damaged erythrocytes (Figure 12A). In some erythrocytes, in the cytoplasm, there were sections through the merozoites of *B. microti* (Figure 11A and Figure 12B). In addition to erythrocytes, nucleated cells—erythroblasts—were observed, which allowed us to conclude that at this stage of development (3 weeks after birth), extramedullary erythropoiesis is still taking place in rats (Figure 12B).

## 4. Discussion

Babesiosis is considered a disease affecting mainly animals, but recently babesiosis has been perceived as a problem also affecting humans. The species with the greatest epidemiological significance for humans are *Babesia microti* and *Babesia divergens*. According to epidemiological reports, the number of cases of tick-borne diseases, i.e., Lyme borreliosis or TBE (tick-borne encephalitis), is increasing year by year in Europe. In the case of babesiosis, the frequency of the disease is also increasing, as well as its geographical range, which is constantly expanding, which is why babesiosis has been included in the group of tick-borne invasive developing diseases [22]. Despite this, there are numerous scientific publications confirming the presence of *Babesia* spp. in ticks, which suggests the possibility of infection with protozoa in many countries, where there is no obligation to report babesiosis infection. The small number of detected cases in, e.g., Poland, may be related to the lack of symptoms or the oligosymptomatic clinical picture of the disease.

The transmission of the disease is associated not only with vector transmission by a tick inhabiting *B. microti* piroplasma but also through the transfusion of blood products, transplantation, or transplacental infection. In 2019, the American Food and Drug Administration (FDA) issued recommendations aimed at reducing the risk of babesiosis infection through blood transfusion, in which it is recommended testing every blood sample taken from donors residing in endemic areas.

In the studies presented in the paper, the most popular animal model in medical research, which is rats, was used. The organs were collected from laboratory rats of the Wistar strain, which were intraperitoneally infected with the reference strain *B. microti* (Franca) Reichenow ATCC 30221. The organs were analyzed under a light and electron microscope. Critical organs most sensitive to *B. microti* infection were selected for the study, i.e., kidneys, responsible for the removal of harmful and toxic metabolic products through which intense blood flow occurs, and the spleen, where abnormal erythrocytes are degraded, including those infected with *Babesia* piroplasms.

In reference animals, the level of parasitemia was controlled by analyzing blood smears stained with the MGG method. This analysis is the gold standard in the diagnosis of babesiosis and should be performed by an experienced diagnostician due to the high similarity of *B. microti* to *P. falciparum*. In the diagnosis of babesiosis, molecular methods are also used, e.g., PCR, IFCA, and ELISA, which are more sensitive than blood smears. *Babesia microti* is a pear-shaped intraerythrocytic protozoan with a complex life cycle with two hosts, i.e., a vertebrate as the intermediate host and a tick as the definitive host.

The clinical picture of babesiosis varies significantly among patients. Symptoms may range from subclinical to severe, with systemic organ failure and death, especially in the presence of HIV infection. The most common symptoms of babesiosis include enlargement of the liver and spleen, congestion of organs, muscle weakness, pain, or jaundice.

The kidney is one of the key organs examined during preclinical studies due to its huge role in the excretion of xenobiotics and metabolic products [23]. In the course of babesiosis, dysfunction of the excretory system may occur. According to the literature, the course of babesiosis is associated with changes in the proximal tubules and, in the later stages of the disease, with mesangial hyperplasia and glomerulonephritis [24]. The literature describes a case of a patient after a splenectomy who developed symptoms of acute renal failure and hemolytic anemia. A blood smear confirmed infection with babesiosis, and parasitemia was about 30%. Histopathological analysis of the kidneys showed damage to the renal tubules and inflammation of the kidneys [25].

In the presented histology, local enlargement of the renal tubules, as well as the space between them, were observed. The blood vessels running between the tubules showed signs of hyperemia. In addition, degenerative changes in the renal parenchyma were visible, as well as the loss of continuity of cell membranes of some tubules and their vacuolation. Vacuolation changes may indicate apoptosis of kidney cells, which occurs as a result of the action of damaging and pathogenic factors, including protozoa. Renal ultrastructure revealed renal corpuscles of normal structure. The renal cortex contained bundles of collagen fibers between the renal tubules, the basement membranes of which were thickened, did not have a homogeneous structure, and their continuity was interrupted in places. The cytoplasm of the cells was diffused with poor osmophilicity. A distorted basal labyrinth was also visible. The impact of *Babesia* spp. infection on the processes leading to kidney dysfunction was described and confirmed in studies conducted by Krawitz, Kavathia, and Choxy [25].

The spleen shows variability in microscopic appearance, especially in aging rats. It should be taken into account that the current changes in the morphology of the organ may result from the direct and indirect toxic effects of many factors. Due to this, the spleen must be carefully examined as a target site for pathogens and to observe the effects of treatment.

A case of a man with *Babesia* merozoites was described in the literature. The spleen parenchyma was soft, brittle, and not enlarged, and the histopathological analysis of the spleen after splenectomy showed changes in the red pulp, in which there was an increase in the number of plasma cells, histiocytes, immunoblasts, and lymphocytes with features of hyperplasia [26].

The histopathological structure of the spleen was also analyzed. Light microscopy showed an indistinct border between the white pulp and the red pulp. The lymph nodules were irregular in shape. In some places, the connective tissue capsule was detached from the organ parenchyma, which could indicate degenerative or edematous changes in the organ. The erythrocytes visible in the cross-section of the blood vessels were multicolored, and the vessels themselves showed signs of hyperemia, and fibrin deposits were visible. In many fields of observation, mitotic cell divisions have been revealed, testifying to the repair and regeneration processes taking place in the structure of the organ. Retrograde changes also occurred in the organ, as evidenced by voids in many areas of the organ. The disintegrating erythrocytes were phagocytized by macrophages. In addition, phagocytosis of *B. microti* protozoa could be observed in many areas.

In the ultrastructure of the organ, the white pulp cells had a variable structure and size. Numerous mitotic divisions were visible in the red matter cells, as well as macrophages with heterophagosomes, in which digested erythrocytes could be observed. During the analysis of slides, there were difficulties in identifying the marginal zones of the spleen. Protozoan trophozoites were present in the erythrocytes and spleen parenchyma cells. Preparations from three-week-old rats contained numerous erythroblasts, which may indicate that extramedullary erythropoiesis was still taking place.

The changes described above are consistent with studies conducted on Mongolian gerbils infected with *B. divergens*, where signs of overactivity, organ enlargement, and blurring of the border between white and red pulp were visible in the collected organs [27].

Different types of immune response mechanisms are involved in the course of infection. During the proliferation of merozoites inside erythrocytes, macrophages and NK cells secrete ROS, TNF-α, and INF-γ, possibly contributing to the decrease in parasitemia [27]. The mechanisms of the humoral response are of little importance in the course of infection with *B. microti*; however, IgG antibodies are responsible for the opsonization of extracellular forms of piroplasms and infected cells. An important role in the response to infection is played by the spleen, which is a link in the mechanism of the cellular immune response [28]. T lymphocytes secrete INF-α causing degradation of intracellular forms of *B. microti*; however, this mechanism is not fully understood [29,30,31]. A lot of unknown facts remain regarding the tropism of different species of the genus *Babesia* to erythrocytes, the diversity of *Babesia* spp. antigens, and information regarding the molecular effects of *Babesia microti* on cells of different organs [32,33].

In the conducted studies, during microscopic observations, both with the use of light and electron microscopy, clusters of T lymphocytes and a large number of macrophages were visible in the spleen. Degenerative changes were also observed, which caused swelling of the organ, which was manifested in the clinical picture of the disease as splenomegaly.

The paper presents the impact of *B. microti* infection on critical organs in the course of infection. The presented results of microscopic observations showed pathological changes in the spleen and kidneys, confirming the impairment of their functions as a result of infection. Accurate knowledge of the mechanism of complications and organ changes in the course of the disease will allow for the improvement of laboratory diagnostic methods, as well as the development of an effective treatment regimen.

## 5. Conclusions

The research presented in this paper led to the following conclusions:Transplacental invasion of *B. microti* causes histological histological as well as ultrastructural changes in the structure of the kidneys and spleen in rats.Changes caused by a transplacental infection in the kidneys and spleen do not limit reproduction, but they show signs of dysfunction of the excretory system.The spleen in three-week-old animals with congenital babesiosis shows typical morphological features and functions characteristically for young rodents, but contact with a pathogen potentially provokes an increased intensity of physiological processes performed by this organ and an increase in its volume.Transplacental infection with *B. microti* does not cause obvious malformations of the observed organs, but microscopic images have indicated that during juvenile life they can cause very serious dysfunctions.The possibility of intrauterine infection of the fetus should encourage the introduction of mandatory screening tests for carriers of pathogenic *Babesia* species, especially in young women, blood donors, and people exposed to tick bites for occupational reasons.

## Figures and Tables

**Figure 1 vetsci-10-00291-f001:**
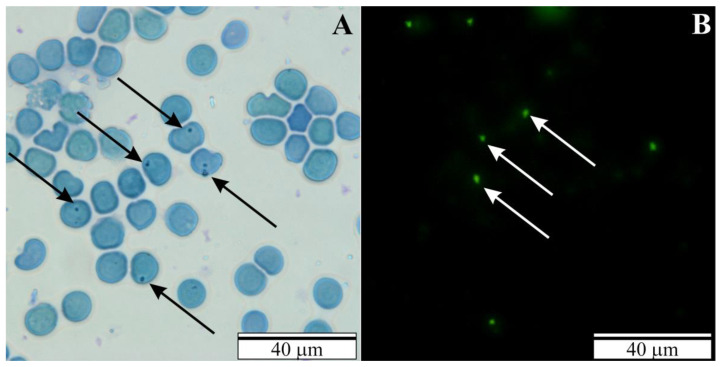
Blood smears from the rats examined with congenital babesiosis. Black arrows—*B. microti* merozoites (**A**) MGG stained and (**B**) FISH stained. White arrows—*B. microti* merozoites.

**Figure 2 vetsci-10-00291-f002:**
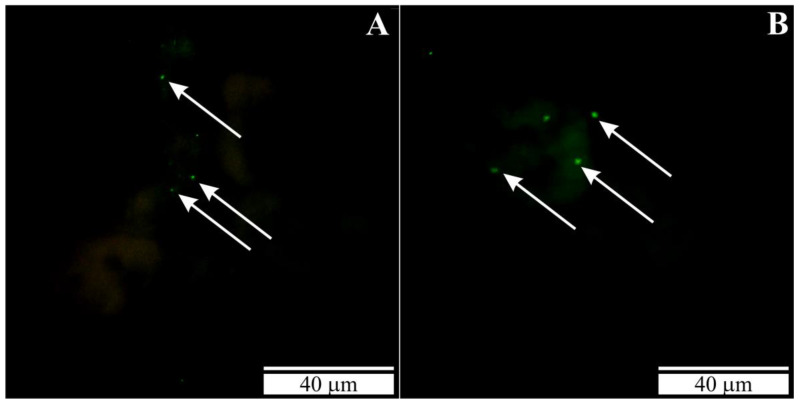
(**A**) Section through the kidney of a rat with congenital babesiosis. (**B**) Section through the spleen of a rat with congenital babesiosis. Slides stained with FISH. White arrows—*B. microti* merozoites.

**Figure 3 vetsci-10-00291-f003:**
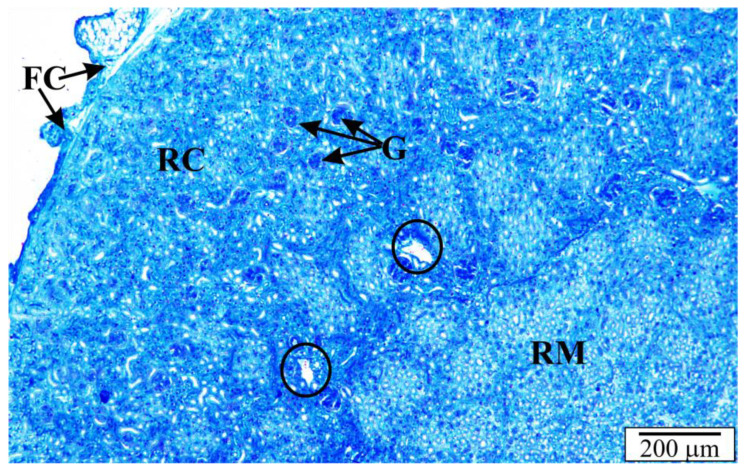
Section through the kidney of a rat with congenital babesiosis. FC—fibrous capsule, G—renal corpuscles, RC—renal cortex, RM—renal medulla, circular markings—extended spaces.

**Figure 4 vetsci-10-00291-f004:**
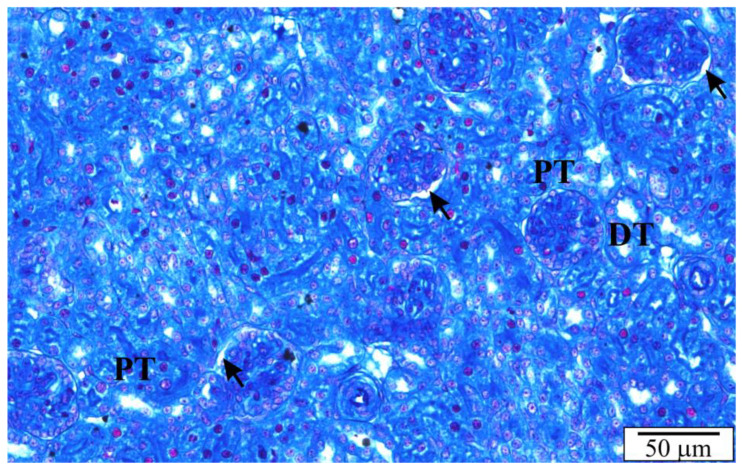
The renal cortex of a rat with congenital babesiosis. DT—distal tubules, PT—proximal tubules, arrows—dilated places of the urinary space of the renal corpuscles.

**Figure 5 vetsci-10-00291-f005:**
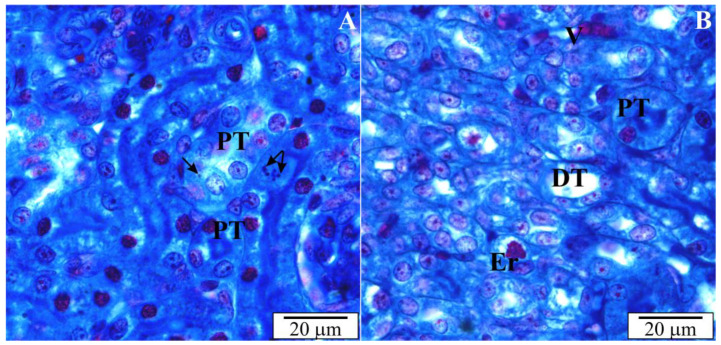
(**A**,**B**) Cross sections through the renal cortex of a rat with congenital babesiosis. DT—distal tubules, Er—erythrocytes inside the intermediate tubule, PT—proximal tubules, V—blood vessel, arrows—*B. microti* merozoites.

**Figure 6 vetsci-10-00291-f006:**
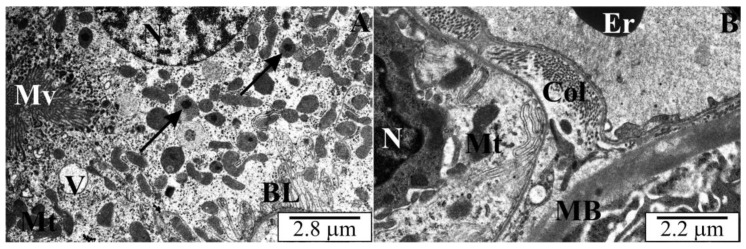
(**A**,**B**) Electronograms of renal cortical sections of rats transplacentally infected with *B. microti*. BL—a basal labyrinth of the proximal tubule, N—cell nuclei proximal tubule, Er—erythrocyte, Col—collagen fibers, MB—basement membrane, Mt—mitochondria, Mv—microvilli, V—vacuole, arrows—peroxisomes.

**Figure 7 vetsci-10-00291-f007:**
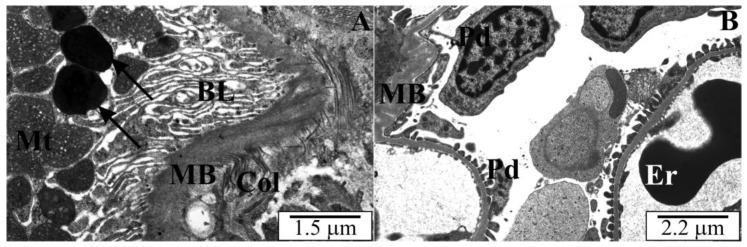
(**A**,**B**) Electronograms of cross sections through the renal cortex of rats with congenital babesiosis. BL—a basal labyrinth, Col—collagen, Er—erythrocyte, MB—basement membrane, Mt—mitochondria, Pd—podocytes, arrows—lysosomes.

**Figure 8 vetsci-10-00291-f008:**
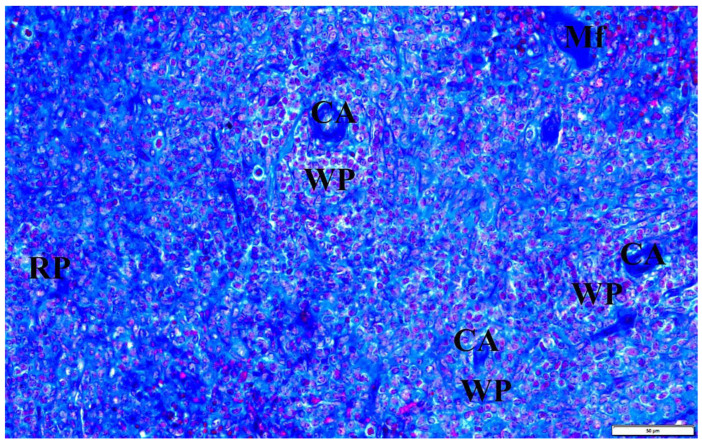
The spleen of a rat with transplacental infection with *B. microti* 3 weeks after birth. CA—central artery, Mf—macrophage, WP—lymphoid follicles (white pulp), RP—red pulp.

**Figure 9 vetsci-10-00291-f009:**
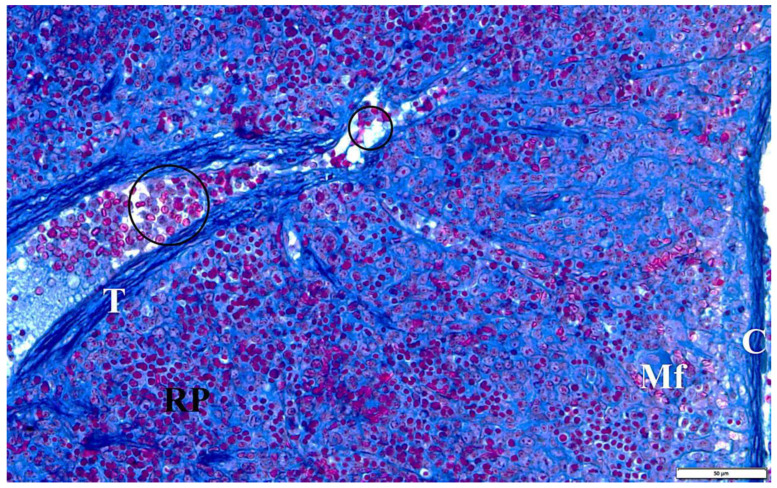
The spleen of a rat with transplacental infection with B. microti 3 weeks after birth. C—fibrous capsule, Mf—macrophage, RP—red pulp, T—trabecula with a trabecular vessel, circular markings—longitudinal sections through trabecular vessels with multicolored erythrocytes and congestive changes.

**Figure 10 vetsci-10-00291-f010:**
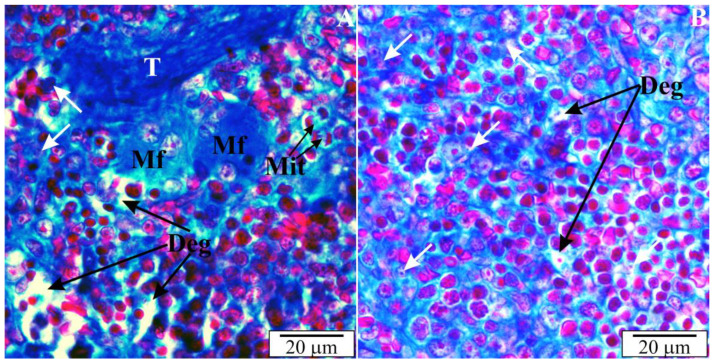
(**A**,**B**) Spleen of a rat with transplacental infection with *B. microti* 3 weeks after birth. Deg—fissured parenchymal dissections (degenerative changes), T—trabeculae, Mf—macrophages, Mit—cells during karyokinesis, white arrows—*B. microti* inclusions.

**Figure 11 vetsci-10-00291-f011:**
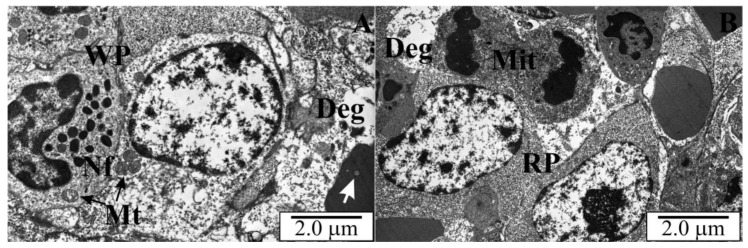
(**A**,**B**) Ultrathin sections across spleens of intrauterine rats infected with *B. microti*. Deg—degenerating zones, Mit—cells during karyokinesis, Mt—mitochondria, Nf—neutrophil, RP—red pulp, WP—white pulp, white arrows—*B. microti* merozoites.

**Figure 12 vetsci-10-00291-f012:**
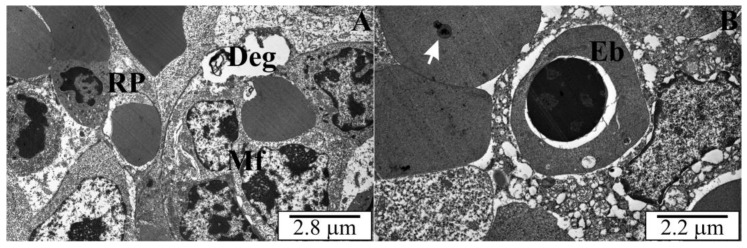
(**A**,**B**) Ultrathin sections across spleens of intrauterine rats infected with B. microti. Deg—degenerating zones, Eb—erythroblast, Mf—macrophage, RP—red pulp, white arrows—B. microti merozoites.

## Data Availability

All the data generated or analyzed in this study are included in this published article.

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
