# Peer review of "Histopathological Analysis of Selected Organs of Rats with Congenital Babesiosis Caused by *Babesia microti"

_vetsci, 2023, doi:10.3390/vetsci10040291_

Round 1
Reviewer 1 Report
The main goal of the present study was to analyze the histopathological lesions observed in the spleen and kidney of young Wistar rats infected trans-placentally with Babesia microti.
Overall, the study is of interest and apparently original in that it is a study in which the interaction of Babesia microti infection is assessed in a transplacentally infected in vivo rat model. The manuscript is in general well written; however, the introduction needs to be expanded and provide additional information extant on the vertical transmission of Babesia microti in the naturally infected species. See for example the following references.
Tolkacz K, Bednarska M, Alsarraf M, Dwuznik D, Grzybek M, Welc-Faleciak R, et al. Prevalence, genetic identity and vertical transmission of Babesia microti in three naturally infected species of vole, Microtus spp. (Cricetidae). Parasit Vectors. 2017;10:66.
Tufts DM, Diuk-Wasser MA. Transplacental transmission of tick-borne Babesia microti in its natural host Peromyscus leucopus. Parasit Vectors. 2018;11:286.
In the introduction, authors should also review the literature available on the use of the mouse model for investigating congenital Babesia microti infection. This information is considered by the reviewer essential to revise so that authors can better justify the use of the rat model in this study. See the following references:
Bednarska M, Bajer A, Drozdowska A, Mierzejewska EJ, Tolkacz K, Welc-Falęciak R. Vertical Transmission of Babesia microti in BALB/c Mice: Preliminary Report. PLoS One. 2015 Sep 15;10(9):e0137731. doi: 10.1371/journal.pone.0137731.
Tołkacz K, Rodo A, Wdowiarska A, Bajer A, Bednarska M. Impact of Babesia microti infection on the initiation and course of pregnancy in BALB/c mice. Parasit Vectors. 2021 Mar 2;14(1):132. doi: 10.1186/s13071-021-04638-0.
In addition, the experimental design of the study conducted in rat model does not seem to be the most adequate as the results presented may not allow authors to conclude that Babesia microti can cause tissue damage in the offspring rat´s kidney and spleen. Authors should also have considered using modern molecular techniques to reliably identify the presence of B. microti in the rats’ tissues. See for example:
Welc-Faleciak R, Bajer A, Bednarska M, Paziewska A, Siński E. Long term monitoring of Babesia microti infection in BALB/c mice using nested PCR. Ann Agric Environ Med. 2007;14(2):287-90.
Shah JS, Mark O, Caoili E, Poruri A, Horowitz RI, Ashbaugh AD, Ramasamy R. A Fluorescence in Situ Hybridization (FISH) Test for Diagnosing Babesiosis. Diagnostics (Basel). 2020 Jun 6;10(6):377. doi: 10.3390/diagnostics10060377.
Finally, but not least important. The conclusions of this study deal with an extrapolation to congenital babesiosis in humans but are not fundamentally based on the results of the experiment conducted in a rat model.
Other comments requiring attention by the authors
Scientific names (B. microti) should be placed in italics.
See lines: 15-16, 19, 26-27, 32, 50, 102, 182, 246, 288, 375, 409, 417, 427, 440, 445, 447.
Introduction.
Line 39. Correct quotation marks in “,,northern malaria”
Line 54. Correct “to e.g. to”
Line 90. Include the parasite infective dose inoculated into the rats' peritoneum
Line 91. ATCC 30221 is a B. microti of mice origin. Apparently not highly pathogenic for rats. Why authors did not use ATCC 30222 instead? which is more virulent, including for hamsters.
Line 107- 136 Section 2.3. Histological and ultrastructural studies
Although considered routine techniques, methodologies for thin and ultrathin sectioning and examination, need references.
Results
Lines 143-148. Please, indicate if all or some of the histopathological structures described are shown in figures 1 to 3.
Line 150. Figure 1 (and for that sake, all of the figures) should be referred to in the text.
Lines 154-155. Legend to figure 2. Arrows are shown in the image. However, there is no indication in the legend as to what structure they are signaling to.
Line 160. Legend to figure 3 indicates arrows signaling to free B. microti merozoites. Are these free merozoites? why they are not inside the erythrocytes present in the blood vessel?. how can authors be sure that these estructures are B. microti merozoites?. Did they perform Fluorescence in Situ Hybridization (FISH) Test?
Lines 167-179. Authors are invited to cite the Figure number in which these ultrastructural observations or changes occur in congenital babesiosis tissues, as compared to tissues processed in the same way but collected from the control group of rats. Authors should indicate what are the comparative differences found both in the microscopical and ultramicroscopic analysis of both types of tissues, kidney, and spleen.
Lines 186-189. Legend to Figure 5. Did authors find any Babesia microti-infected erythrocytes within the kidneys' blood vessels/capillaries?
Lines 218-221. Legend to Figure 7. Did authors find any Babesia microti-infected erythrocytes within the spleen intratrabecular blood vessels with multicolored erythrocytes?
Line 227-228. Please indicate the figure in which “phagocytosis of disintegrating erythrocytes and B. microti protozoa” are depicted.
Lines 230-233. Legend to Figure 8. Please indicate host cells where “B. microti inclusions” are presumably observed.
Line 246. Legend to Figure 10. It is mentioned, “rats infested intrauterine with B. microti.”. The correct term is “infected”, instead of “infested”.
Lines 249- 250. Authors should indicate the figure number to which they are referring with the presence of “free trophozoites”. This is so that the readers have a better appraisal.
Lines 259-261. Legend to Figure 11. What is the size of the B. microti merozoite or trophozoite? The morphology of the inclusion shown does not look like a classical Intraerytrocitic Babesia structure! In Fig 11 B. Is the inclusion present in the erythroblast stroma part of a B. microti parasite?
Line 267. Spell out, first-time use of the acronym "OZM"
Line 276. Do authors refer to “inhabiting” as “hosting”?
Lines 277-280. The paragraph needs a reference.
Lines 281-283. Authors should discuss why they did not use mice, more similar to the natural host, or else Hamsters, more susceptible to B. microti infection. In addition, the Authors should be more specific in terms of the organs collected. If rats were sacrificed, why did not collect uteri, placenta, or fetuses? It is not clear if the authors collected organs from the pregnant rats or from the offspring rats.
Line 289. Please check the statement “In reference animals, the level of parasitemia was controlled by analyzing blood smears”, as it is written, it is inferred that analysis of blood smears conducts to control Babesia parasitemias
Line 290. “MGG”. Define the staining method.
Line 292-293. Why authors did not use any of those more sensitive methods mentioned?
Lines 296-299. Did the authors find any of these symptoms? (rather clinical signs in the studied rats).
Lines 303-308. Did the authors find any of those changes described in the paragraph, in the studied rats? Is the rat a good model for congenital B. microti infection? Authors should discuss this issue.
Lines 319-321. Did rats infected with B. microti in this study show renal dysfunction based on clinical pathology evidence?
Line 327. “A case of a man with Babesia merozoites” found in the spleen?
Lines 347-348. “Protozoan trophozoites were present in erythrocytes and spleen parenchyma cells”. Is this clearly shown in the spleen thin and ultrastructure images presented? It is not perfectly clear if the histopathology and ultrastructure images are from the 3-week-old rats or from the adult female rats that were intraperitoneally inoculated with B. microti infected erythrocytes?
Lines 370-372. Authors claim that “Accurate knowledge of the mechanism of complications and organ changes in the course of the disease will allow for the improvement of laboratory diagnostic methods, as well as the development of an effective treatment regimen” Authors are invited to cite more precise examples on the improvement of diagnostic methods and development of effective treatment regimen.
Lines 377-378. Do the authors refer to changes in the spleen and kidney of the female pregnant rats that were infected with B. microti? Or to tissue changes in the offspring rats born from intraperitoneally infected rats?
Lines 383-385. No such serious dysfunctions were analyzed/described in this study
Lines 386-389. Can this conclusion be arrived at and proposed to be mandatory in people, based on a rat study with doubtful B. microti infectious status?.
References
Line 412. Delete “PMID: 28202022; PMCID: PMC5310009”
Line 416. Delete “PMID: 34578196; PMCID: PMC8468516..”
Line 419-420. “Fabiani S., Fortunato S., Bruschi F. Solid Organ Transplant and Parasitic Diseases: A Review of the Clinical Cases in the Last Two Decades. Pathogens 2018; 7(3):65.” Should be numbered as reference 6.
Line 421. Reference 6 should now be reference 7 and subsequent references should also change the number in the references list. Check correspondence with citation numbers in the text of the manuscript. Check also correct citation numbering once additional references are included in the text.
Line 422. Delte “PMID: 34993053; PMCID: PMC8713127.”
Line 426. Delete “PMID: 26629450; PMCID: PMC4557163.”
Line 438. Delete “the official journal of the National Kidney Foundation”
Line 442. Delete “PMID: 34880327; PMCID: PMC8654915.”
Line 446. Delete “PMID: 29445365; PMCID: PMC5797759.”
Author Response
Dear Reviewer
The authors would like to thank you for Your valuable opinions. Below we have described (in red font) the corrections made to the manuscript and provided our opinions on some of the problems.
In the Introduction chapter, the authors supplemented required information, regarding studies on the transplacental transmission of Babesia microti in various rodents, including Peromyscus leucopus, M. arvalis, M. oeconomus, and BALB/c Mice.
The course of experimental infection with B. microti (Franca) strain Reichenow ATCC 30221 in laboratory rats has been studied in our Department for many years. We found that Wistar rats succumbed to infection with this B. microti strain, which caused quite many changes in the organs of these animals [Okła H, Jasik KP, Słodki J, Rozwadowska B, Słodki A, Jurzak M, Pierzchała E. Hepatic tissue changes in rats due to chronic invasion of Babesia microti. Folia Biol (Krakow). 2014;62(4):353-9. doi: 10.3409/fb62_4.353.; Okła Hubert, Jasik Krzysztof P., Urbańska-Jasik Danuta, Słodki Jan, Rozwadowska Beata, Grelowski Michał, Chmielik Ewa, Słodki Aleksandra [et al.] (2017). Rat spleen in the course of Babesia microti invasion: histological and submicroscopic studies. "Acta Protozoologica" (2017, vol. 56, pp. 129-137), doi 10.4467/16890027AP.17.011.7486; Albertyńska M, Okła H, Jasik K, Urbańska-Jasik D, Pol P. Interactions between Babesia microti merozoites and rat kidney cells in a short-term in vitro culture and animal model. Sci Rep. 2021 Dec 8;11(1):23663. doi: 10.1038/s41598-021-03079-0.].
We also observed a significant prevalence of vertical transmission in rats of the pathogens used in the experiments [Krzysztof P. Jasik, Hubert Okła, Jan Słodki, Beata Rozwadowska, Aleksandra Słodki, and Weronika Rupik. Congenital Tick-Borne Diseases: Is This An Alternative Route of Transmission of Tick-Borne Pathogens In Mammals? Vector-Borne and Zoonotic Diseases.Nov 2015.637-644.http://doi.org/10.1089/vbz.2015.1815; Albertyńska M, Rupik W, Hermyt M, Okła H, Jasik KP (2017) Babesia Microti – Known and Unknown Protists. Glob J Zool 2(1): 001-007. DOI: 10.17352/gjz.000004.].
The authors did not want to multiply the documentation and therefore omitted parasitemia detection in the tested rats. However, both traditional analyses of blood smears, stained by the May-Grünwald-Giemsa method, and a FISH test were performed to confirm the presence of piroplasms in the blood and tissues of the test animals. Documentation of these initial analyses was attached to the manuscript.
Of course, the rat studies represent a stage of preliminary in vivo research. Humans are accidental hosts for Babesia microti, however not uncommon. This is indicated by clinical data, described mainly in the US.
The reviewer points out that the rat is not a typical host of Babesia - in the Authors' opinion, this animal model is then an even more appropriate example for studying infection in humans, in whom Babesia is also an atypical parasite.
Considering the severe changes in the spleen and kidneys of Babesia microti - infected rats that we have described, we thought it worthwhile to extend this experiment to the analysis of transplacental-infected animals. We would like to analyze human biopsy specimens with hematological and nephrological dysfunctions soon, taking into account the possibility of babesiosis in such patients.
We have corrected all the comments indicated below in the manuscript.
The reviewer points out that the rat is not a typical host of Babesia - in the Authors' opinion, this animal model is then an even more appropriate example for studying infection in humans, in whom Babesia is also an atypical parasite. Most importantly, the Babesia microti strain used in the study was originally isolated from humans. It is therefore potentially invasive to humans.
The results obtained were compared with the control group. however, we did not include images with normal tissues, because we did not want to multiply the documentation additionally.
We hope that the modifications made to the manuscript and our responses to the reviewers will be accepted.
Best regards
Krzysztof Jasik
Reviewer 2 Report
Dear authors,
Nice work done. I have some comments.
1. the aim of the study is not presented clearly.
2. the parasitemia observed means clinical disease? Did you find compatible clinical signs?
3. How many of the 4 rats developed these microscopic alterations? How many repetitions per examined tissue did you perform?
4. I think that some kind of statistical analysis should be applied.
5. I recommend the authors to make an extensive comparison of their lessions with the normal tissues from the control group. Did they make such approach.
Thanks in advance!
Author Response
Dear Reviewer
The authors would like to thank you for Your valuable opinions. Below we have described (in red font) the corrections made to the manuscript and provided our opinions on some of the problems.
1. The clinical manifestations of babesiosis are very unclear. In our previously published papers [Rat spleen in the course of Babesia microti invasion: histological and submicroscopic studies. "Acta Protozoologica" (2017, vol. 56, pp. 129-137), doi 10.4467/16890027AP.17.011.7486; Albertyńska M, Okła H, Jasik K, Urbańska-Jasik D, Pol P. Interactions between Babesia microti merozoites and rat kidney cells in a short-term in vitro culture and animal model. Sci Rep. 2021 Dec 8;11(1):23663. doi: 10.1038/s41598-021-03079-0.], we showed clear histopathological changes in the spleen and kidneys. In the submitted manuscript, we would like to present the results of analyses of the effects of congenital babesiosis on these organs. We have more precisely formulated this goal in the text of the manuscript, as the last sentence of the INTRODUCTION.
2. The clinical manifestations of babesiosis are very unclarified, so there is a need for a broader analysis of the effects of babesiosis, including histopathological aspects.
3. For this study, only the transplacental-infected animals were used. This was verified by analyzing blood smears and by FISH. As a correction, these data were included in the manuscript.
4. In basic histological studies, we cannot introduce statistical tests. We do not have any numerical data.
The results obtained were compared with the control group. however, we did not include images with normal tissues, because we did not want to multiply the documentation additionally.
We hope that the modifications made to the manuscript and our responses to the reviewers will be accepted.
Best regards
Krzysztof Jasik
Round 2
Reviewer 1 Report
The main goal of the present study was to analyze the histopathological lesions observed in the spleen and kidney of young Wistar rats infected trans-placentally with Babesia microti.
Overall, the study is of interest and apparently original in that it is an study in which the interaction of Babesia microti infection is assessed in a transplacentally infected in vivo rat model. The manuscript is in general well written, however, moderate english changes are required for better reading.
Author Response
Dear Reviewer
The authors thank you for your valuable comments. As suggested, I have revised the correctness of the English language (US). I have made minor corrections and forwarded the text to a native speaker employed at the Medical University of Silesia for checking.
If the Editor and Reviewers feel that there are still ambiguities in the text, please feel free to point out paragraphs that should be corrected.
I am sending the corrected text in a pdf version and a Word version (track changes) so that my corrections are visible.
Kind regards
Krzysztof Jasik

Reviewer 2 Report
Thanks for your honest answers. My comments have been addressed.
Author Response

(The authors gave the same response as above.)
